systems biology/biomathematics/applied mathematics

bifurcation, stability analysis, parameter inference

**Author for correspondence:**
Michael P. H. Stumpf
e-mail: mstumpf@unimelb.edu.au

# Parameter inference in dynamical systems with co-dimension 1 bifurcations

## Elisabeth Roesch and Michael P. H. Stumpf

Melbourne Integrative Genomics, School of BioScience and School of Mathematics & Statistics, University of Melbourne, Parkville, Australia

MPHS, 0000-0002-3577-1222

Dynamical systems with intricate behaviour are all-pervasive in biology. Many of the most interesting biological processes indicate the presence of bifurcations, i.e. phenomena where a small change in a system parameter causes qualitatively different behaviour. Bifurcation theory has become a rich field of research in its own right and evaluating the bifurcation behaviour of a given dynamical system can be challenging. An even greater challenge, however, is to learn the bifurcation structure of dynamical systems from data, where the precise model structure is not known. Here, we study one aspects of this problem: the practical implications that the presence of bifurcations has on our ability to infer model parameters and initial conditions from empirical data; we focus on the canonical co-dimension 1 bifurcations and provide a comprehensive analysis of how dynamics, and our ability to infer kinetic parameters are linked. The picture thus emerging is surprisingly nuanced and suggests that identification of the qualitative dynamics—the bifurcation diagram—should precede any attempt at inferring kinetic parameters.

## 1. Introduction

Modelling in the physical sciences often proceeds in a rigorous and disciplined manner: a small set of principles is sufficient to develop theoretical models of e.g. molecular dynamics, transport processes in solids or transitions between different phases in condensed matter theory [1]. Symmetries then lead to conservation laws which can guide model development and greatly add in the interpretation of such models [2].

However, in many domains, including biology, modelling has to follow a different procedure [3]: for example, if the basic symmetries are too far removed from the processes that we want to model, or the system is too complex (or indeed complicated) to model based on *first principles* [4]. In these scenarios, we typically have to develop models based on domain expertise and subsequently compare them with available data using e.g.

rigorous statistical model selection or model checking methods [5]. There is, as a result, a vast literature on *reverse engineering* and *inverse problems* [6–12]. These sets of approaches allow us to develop models—typically iteratively—in light of available data and background information: we define the models, design more discriminatory experiments, make testable predictions about the behaviour of complex systems, and gain mechanistic insights into the inner workings of such systems. Reverse engineering and statistical model selection methods have found widespread use in many disciplines, ranging from engineering and biology to economics and the social sciences [10].

Much of the literature in this area is focused on important systems where dynamics can be described or at least approximated in terms of linear differential equation models [8,13]. For nonlinear and stochastic dynamics, problems exacerbate very quickly. Here, we address one aspect of reverse engineering that is of particular importance in many applied sciences. *Bifurcations* in dynamical systems result in a qualitative change in system behaviour; they are important in a host of biological processes, ranging from cell cycle control [14], cell fate decision making [15,16], to ecological [17] and epidemiological problems [18], and neurophysiology [19]. In developmental biology, for example, we often seek to identify the stable stationary fixed points of an ordinary differential equation (ODE) system with distinct *cell states* [15,20,21]. Equally, when considering an infectious disease, we also distinguish different stationary points of the population dynamics with different meaning, e.g. disease is controlled versus disease has spread. In modelling such systems, we want to be able to capture the qualitative dynamics accurately. And this means that we have to be able to detect qualitative change points or bifurcations.

Here, we focus on the simplest bifurcations in their simplest setting: scalar one-dimensional versions of four canonical types of bifurcations [22]. We investigate how the four bifurcations effect our abilities in inverse engineering, more specifically *parameter inference*. Similarly to Kirk *et al.* [23], we systematically investigate how the shape of the log-likelihood function of parameters change while the system undergoes a bifurcation. Here, we consider the class of co-dimension 1 bifurcations, which are of profound importance in biology; understanding the factors that allow us to learn the structure and parameters of such bifurcating systems in the setting is an important prerequisite for recognizing these qualitative change points in higher dimensions.

## 2. Methods

### 2.1. Bifurcations

In this study, we investigate dynamical systems modelled by ODEs that undergo *bifurcations*. At bifurcation points a system's behaviour may differ qualitatively depending on small changes in parameters [22]. Previous work has shown that the existence of bifurcations can profoundly affect our ability to infer parameters [23,24] and here we extend this by considering and contrasting the canonical *co-dimension 1* bifurcations. These are bifurcations where the qualitative change is caused by the variation in a single parameter, which we refer to as the *bifurcation parameter*, $\alpha$. We look at them in their simplest normal forms which are defined via the following ODEs:

*Saddle-node* bifurcation

$$\frac{\partial X}{\partial t} = \alpha - X^2, \tag{2.1}$$

*Transcritical* bifurcation

$$\frac{\partial X}{\partial t} = \alpha X - X^2, \tag{2.2}$$

*Supercritical pitchfork* bifurcation

$$\frac{\partial X}{\partial t} = \alpha X - X^3, \tag{2.3}$$

*Subcritical pitchfork* bifurcation

$$\frac{\partial X}{\partial t} = \alpha X + X^3, \tag{2.4}$$

where $X$ is the observable species and $t$ is the time. Figure 1 depicts the bifurcation diagrams of these systems.

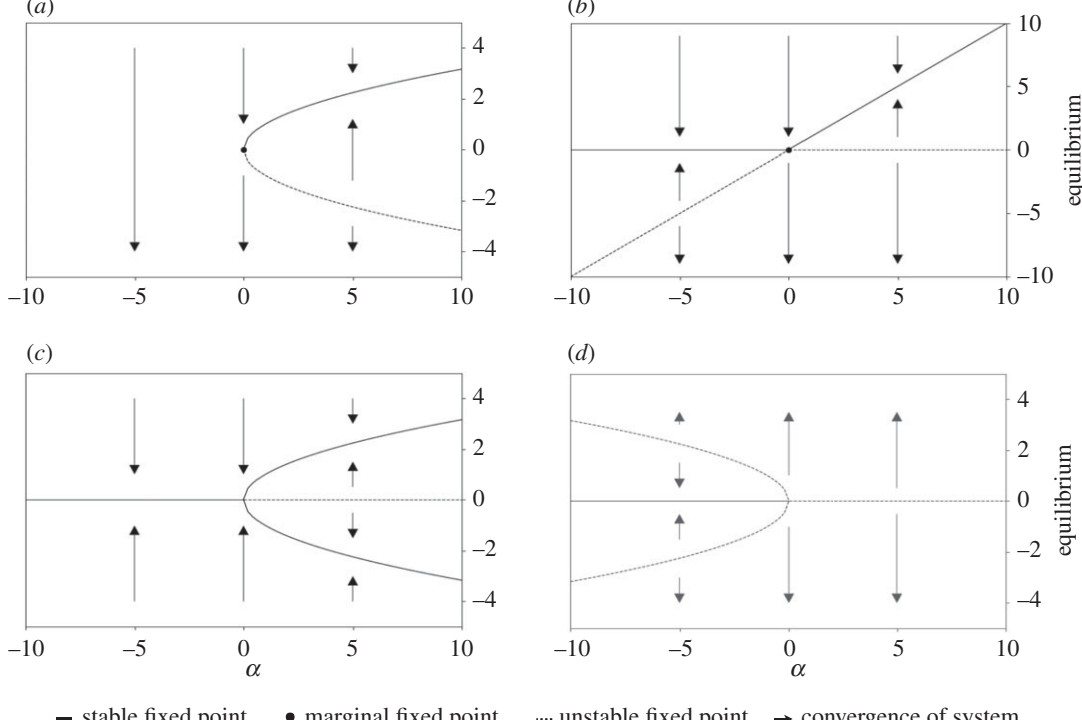

— stable fixed point    • marginal fixed point    ···· unstable fixed point    → convergence of system

**Figure 1.** *Bifurcation diagrams for the saddle-node, transcritical and super/sub-critical pitchfork co-dimension 1 bifurcation.* The diagrams visualize the global stability properties of dynamical systems (equations (2.1)–(2.4)) depending on the bifurcation parameter $\alpha$ and the initial condition of the system (IC). We see that due to small, smooth changes in the parameter space, the system's behaviour changes in a qualitative manner as fixed points appear, disappear or change their stability properties. The graph represents fixed points which are stable (solid line) or unstable (dashed line) and arrows describe the convergence or divergence to/from a fixed point of a given system. If a fixed point is marginal it holds stable and unstable stability properties (dot). (*a*) Saddle-node bifurcation, (*b*) transcritical bifurcation, (*c*) supercritical pitchfork bifurcation and (*d*) subcritical pitchfork bifurcation.

In the first system, a saddle-node bifurcation occurs at $\alpha = 0$ (figure 1*a*, saddle-node bifurcation). For $\alpha < 0$, no fixed points exist and the system diverges whereas for $\alpha > 0$ one stable and one unstable fixed point exist ($X_{\text{stable}} = \sqrt{\alpha}$, $X_{\text{unstable}} = -\sqrt{\alpha}$). The second system undergoes a transcritical bifurcation (figure 1*b*, transcritical bifurcation). Again, we observe the qualitative stability change at $\alpha = 0$. For $\alpha < 0$ and for $\alpha > 0$ two fixed points exist; however, their locations vary ($\alpha < 0$: $X_{\text{unstable}} = -\alpha$ and $X_{\text{stable}} = 0$, $\alpha > 0$: $X_{\text{stable}} = \alpha$ and $X_{\text{unstable}} = 0$).

The third and fourth systems both exhibit pitchfork bifurcations representing the supercritical and subcritical case, respectively (figure 1*c*, supercritical pitchfork bifurcation and figure 1*d*, subcritical pitchfork bifurcation). The system behaviour is symmetric along $X = 0$ for both types. However, the positioning of the fixed points and their stability properties of the fixed points are reversed (supercritical: $\alpha <= 0$: $X_{\text{stable}} = 0$, $\alpha > 0$: $X_{\text{stable}} = \pm\sqrt{\alpha}$ and $X_{\text{unstable}} = 0$; and subcritical: $\alpha <= 0$: $X_{\text{unstable}} = \pm\sqrt{\alpha}$ and $X_{\text{stable}} = 0$, $\alpha > 0$: $X_{\text{unstable}} = 0$).

## 2.2. Likelihood

We investigate how the inference of parameters, but also of initial conditions, is affected by qualitative changes in these systems. In particular, we are trying to understand how the likelihood [25] over parameters and initial conditions is affected by the presence of a bifurcation. We follow the general approach of Kirk *et al.* [23]. The log-likelihood [25] for such problems (assuming identically and independently normally distributed experimental noise) can be written, up to a constant proportionality factor, as

$$\ln\left(L(\theta|D)\right) \propto -\frac{1}{2}\sum_{n=1}^{M}(d(y_i, \mu_i(\theta))^2), \tag{2.5}$$

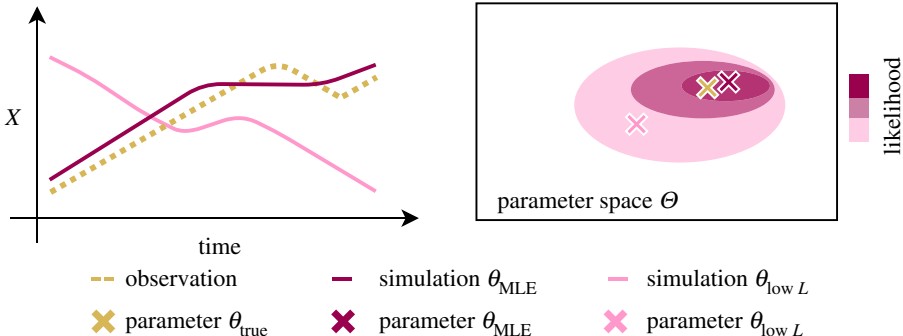

**Figure 2.** *Intuition for the likelihood function.* The likelihood of a parameter combination $\theta$ expresses how well the simulation of $\theta$ describes the observation $D$. Therefore, the simulation of the model with a parameter combination with high likelihood such as $\theta_{MLE}$ represents the observed data well whereas the model simulation of the parameter combination $\theta_{low\,L}$ differs significantly from the observation. Therefore, the true parameter combination $\theta_{true}$ is assumed to be closer to $\theta_{MLE}$ than $\theta_{low\,L}$ in the parameter space $\Theta$.

where $\theta = (\alpha,\ \mathrm{IC})$ is the combined vector of rate parameter and initial condition; $D = [y_1,\ y_2,\ ...,\ y_M]$ denotes the observations; and $[\mu_1(\theta),\ \mu_2(\theta),\ ...,\ \mu_M(\theta)]$ represents the theoretical outputs for given $\theta$; and where $d$ is the Euclidean distance between the theoretical and observed data.

The value of $\theta$ for which equation (2.5) becomes maximal is the *maximum-likelihood estimate* (MLE) [25] and can be obtained numerically. Here, as we are interested in assessing how inference is affected by such changes, we calculate the likelihood over the $\alpha$,IC plane. An intuitive explanation for the likelihood is given in figure 2.

## 2.3. Fisher information

The MLE denotes the most likely value of the parameter given the observed data. However, it does not give an assessment of how much it is better compared with a different value of the parameter. The Fisher information [23,25,26] gives a more nuanced view of how reliably we can infer parameters, or how much certainty we should have in a given inferred parameter value: it is a measure of how quickly the likelihood changes around the MLE and needs to be understood as the curvature of the log-likelihood function. The *observed* Fisher information $\mathrm{FI}_{obs}$ is given by the second derivative (or the Hessian matrix in the multivariate case),

$$\mathrm{FI}_{obs}(\theta) = -(\mathrm{Hessian}(\ln(L(\theta|D)))). \tag{2.6}$$

The entries, $H_{ij}$ of the Hessian matrix, $H$, provide the curvatures,

$$\mathrm{FI}_{ij} = -H_{ij} = -\frac{\partial^2 \ln(L(\theta|D))}{\partial\theta_i\partial\theta_j}, \tag{2.7}$$

Intuitively, a more peaked log-likelihood surface around the MLE will have a higher Fisher information than a less peaked surface: this is because for the former, even a small shift away from the MLE will result in a substantial decrease of the log-likelihood, compared with the latter. The Fisher information quantifies this and is hence a measure of how well a parameter is inferred.

We summarize the Fisher information matrix (as our preferred measure of inferability) by taking the trace of the matrix

$$\mathrm{FI}_{obs}(\theta) = -\mathrm{tr}(\mathrm{Hessian}(\ln(L(\theta|D)))) = -\sum_i H_{ii}. \tag{2.8}$$

This value summarizes the information content that the observations $D$ carry about the bifurcation parameter $\alpha$ and the initial condition IC. An illustration is given in figure 3. The Fisher information is a local property around the MLE (or *maximum a posteriori* (MAP) estimate) and as such has limitations in many practical applications; in practice, however, it can still provide useful guidance [27] about parameter inferability.

## 2.4. Generating simulated data

In this work, we rely purely on simulated data. This means that we generate simulations of the system and treat them as the observation $D$. The advantage of this is that we are able to easily vary the level of difficulty of the problem and, most importantly, rigorously test the inference procedure. In the first part of

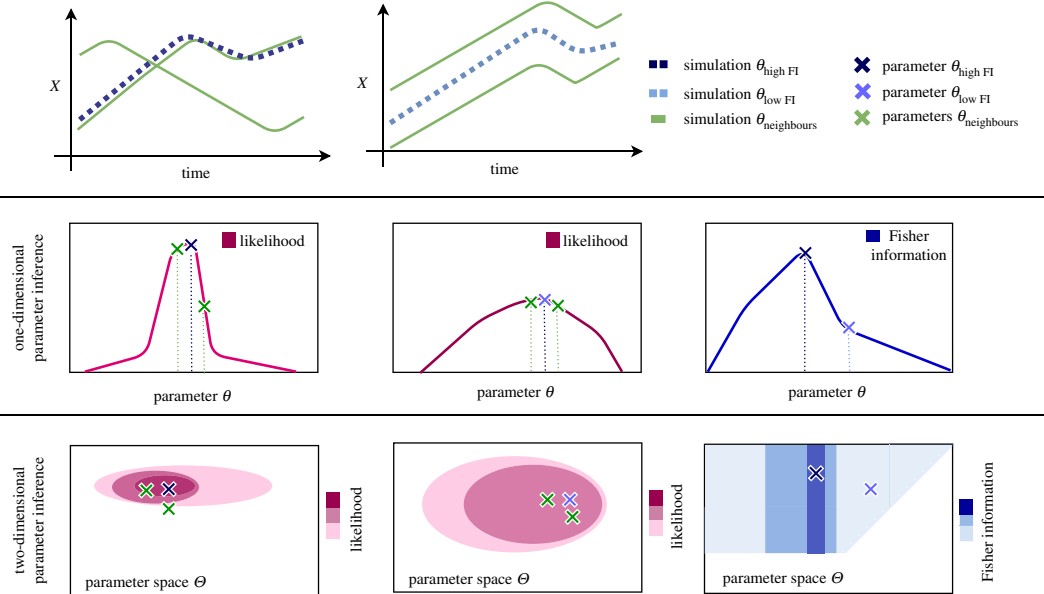

**Figure 3.** *Intuition for the Fisher information.* For a parameter combination with high Fisher information $\theta_{\text{high FI}}$ (the MLE of the likelihood function of $\theta_{\text{high FI}}$) neighbouring parameter samples may show significantly different likelihood values. If this is the case the distance of the simulations of the neighbouring parameters to the observation, and subsequently to the simulation of $\theta_{\text{high FI}}$, is large. Neighbouring parameter samples of $\theta_{\text{low FI}}$ (the MLE of the likelihood function for $\theta_{\text{low FI}}$), however, result in more similar values. Therefore, the simulations of neighbouring parameter are equally distant to the observation and the simulation of $\theta_{\text{low FI}}$. In this figure, we show the one-dimensional case (e.g. $\theta = \alpha$) and the two-dimensional case (e.g. $\theta = (\alpha, \text{IC})$). In the one-dimensional case, the blue line graph repesents the Fisher information and indicates for each value of $\theta$ the observed curvatures of its one-dimensional likelihood function. In the two-dimensional case, the Fisher information is presented via a contour plot where intense/weak colour indicates high/low Fisher information. The two-dimensional likelihood surface of a parameter combination with high Fisher information $\theta_{\text{high FI}}$ has a peaky shape (high curvature) whereas a parameter combination with low Fisher information $\theta_{\text{low FI}}$ results in a broader shape (low curvature).

the results, we focus on the ideal scenario. In this case, the observation $D$ is the exact ODE solution calculated with the fourth-order Runge–Kutta method and $D$ contains observations at 1000 time points (for more details see the code linked in the Data accessibility section). Subsequently, we investigate the effect of experimental noise and sparser data on the parameter inference performance.

### 2.4.1. Noise

Due to the imperfect nature of experimental data, we know that measurements of the observation $D$ are afflicted by noise. To assess the effects of such noise on parameter inference, we generate noisy data and investigate how the likelihood changes with increasing noise for the same parameter combination. The noise term follows a zero-centred *Gaussian* distribution $\mathcal{N}$ with standard deviation $\sigma$,

$$\text{noise} \sim \mathcal{N}(0, \sigma^2), \tag{2.9}$$

and noise levels are defined as low ($\sigma = 0$), moderate ($\sigma = 0.1$) and heavy ($\sigma = 1.0$); other noise models, e.g. log-normal are also possible, of course.

### 2.4.2. Sparsity

To better reflect many real-world applications we also consider the case of fewer data points, $M$, and the impact that this has for inference (which was also discussed at length in [23]). We consider very high ($M = 1000$), high ($M = 500$) and low ($M = 10$) sampling rates.

## 3. Results

We first present the results for the saddle-node bifurcation, before summarizing the results for the three other bifurcations. Finally, we discuss the effects of sparse and noisy data on the parameter inference problem.

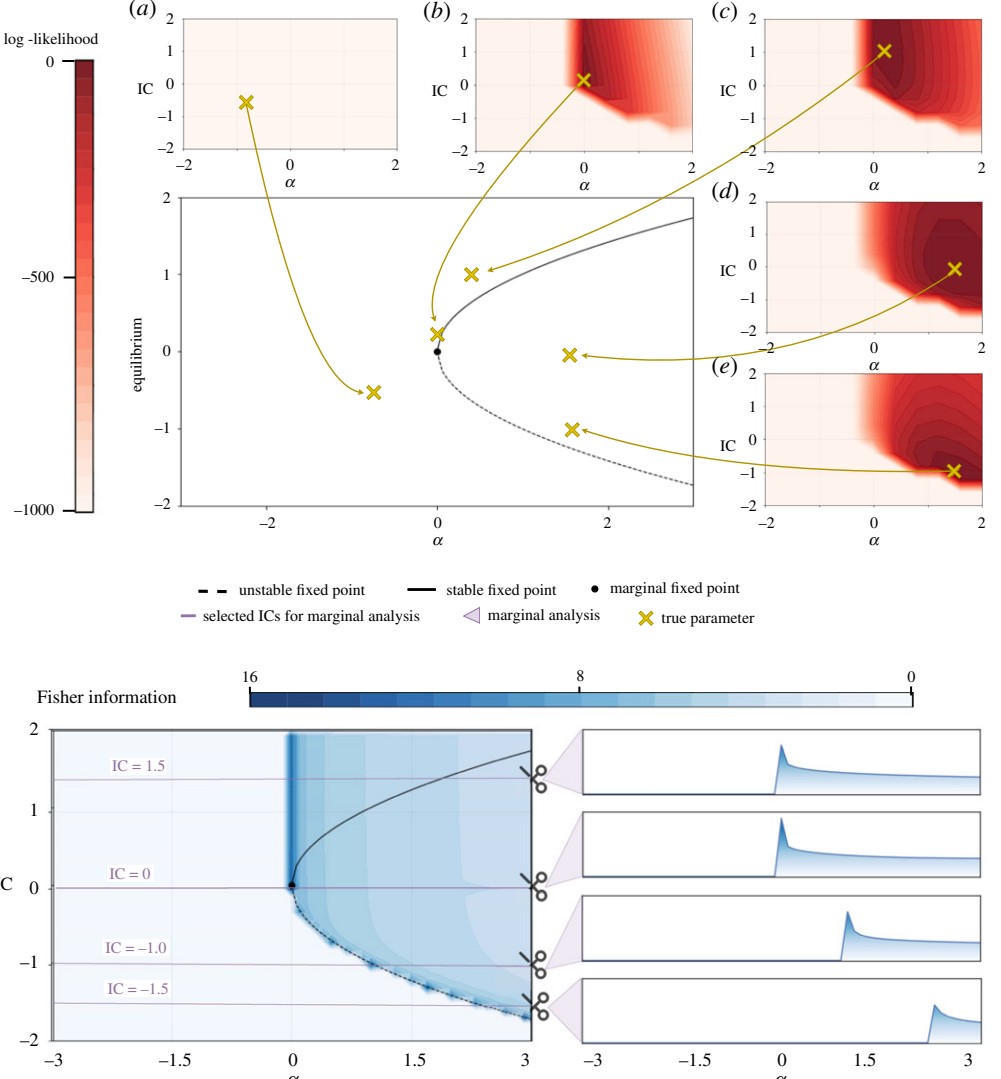

**Figure 4.** *Quantifying information content around the saddle-node bifurcation via the log-likelihood function and Fisher information. In red, the log-likelihood surfaces of selected parameter combinations are depicted. They are surrounding the bifurcation diagram of the saddle-node bifurcation in order to set the selection of parameters in global stability context. Parameter combinations with (a) negative $\alpha$ give rise to diverging solutions. The likelihood function of a parameter combination close to the bifurcation event at $\alpha = 0$ but on the non-negative side (b) shows a different behaviour and a pronounced peak can be observed. This is due to even minor changes of $\alpha$ causing the stability of the whole system to change. Therefore, parameter combinations can be inferred precisely in this area. Moving away from the bifurcation event but for $\alpha > 0$ (c,d), we observe broader likelihood shapes. This means parameters can be inferred, but the certainty decreases. Staying in the case where $\alpha > 0$, but looking at parameter combinations closer to the unstable fixed point $X_{\text{unstable}} = -\sqrt{\alpha}$ (e), the likelihood surfaces have more pronounced peaks again. This is due to the fact that the slightest change of IC to smaller values causes the system to diverge. In blue, the Fisher information is calculated for the whole parameter space. Each data point $(\alpha_x, IC_y)$ represents the observed Fisher information $FI_{\text{obs}}(\theta_{x,y})$ where $\theta_{x,y} = (\alpha_x, IC_y)$. This value summarizes the curvature of the log-likelihood for the parameter combination and we use it to measure the inferability of the parameters. For four IC, the marginalized Fisher information with increasing $\alpha$ is presented.*

## 3.1. Likelihood estimation around the saddle-node bifurcation

The parameters we wish to estimate are the bifurcation parameter, $\alpha$, and the initial condition of the system, IC. In figure 4, we select a group of exemplary parameter combinations covering the possible qualitative system behaviour and display the associated likelihood surfaces: no fixed point ($\theta_a$), single marginal fixed point ($\theta_b$)), and systems with one stable and one unstable fixed point ($\theta_c$–$\theta_e$).

The first parameter combination, $\theta_a$, represents combinations of negative $\alpha$ and arbitrary IC. As there are no fixed points in this region of the parameter space, the systems do not converge to a stationary

point but drift off to $-\infty$. Parameter and initial condition inference become essentially impossible as the log-likelihood decreases rapidly along these diverging system trajectories.

When $\alpha$ and IC become positive, the log-likelihood surface changes drastically. As an example, we show the likelihood surface of $\theta_b$, which is now very close to the bifurcation event at $\alpha = 0$ and fulfils $\alpha$, IC >= 0. We can identify a pronounced peak in the log-likelihood surface at the location of the true parameter combination, and accurate parameter inference becomes possible in this domain; around $\alpha = 0$, even slight changes in the parameter $\alpha$ can result in qualitatively different system dynamics. In the case of $\theta_b$, solutions with even infinitesimally smaller $\alpha$ will diverge, and with the likelihood surface, we can rule such values out. A similar situation is observed for different initial conditions, as ICs below the stationary point result in diverging solutions.

For the final category of parameter combinations, we increase $\alpha$ further. Here, two fixed points exist, where one is stable and one is unstable. The parameter combination $\theta_c$ is located above the stable fixed point, whereas $\theta_d$ and $\theta_e$ are between the two fixed points. For all three parameter combinations, we can estimate the true parameter from the peak of the likelihood surface which does indeed cover the true parameter combination. Also, like in the last scenario, the log-likelihood readily rules out any parameter combinations with negative $\alpha$ or IC. However, we observe that from $\theta_b$ to $\theta_c$ to $\theta_d$ the shape of log-likelihood surface broadens, i.e. inference becomes harder as we move away from the bifurcation event towards larger positive values of $\alpha$. Initial conditions close to the unstable fixed points result in a more pronounced peak of the likelihood surface, as initial conditions below the unstable fixed points can be ruled out by the likelihood.

## 3.2. Fisher information around the saddle-node bifurcation

Looking at the exemplar likelihood surfaces in parameter space gives us a good idea of the general ability to perform parameter inference around the bifurcation point. It allows us to investigate the overall shape of the log-likelihood and it enables us to assess how successful or straightforward parameter inference on different sides of the bifurcation will be. For example, in figure 4, the log-likelihood surfaces of $\theta_b$ and $\theta_d$ both cover the true parameter. However, the quality of the two-parameter estimates differs as the log-likelihood of $\theta_b$ is more peaked than the log-likelihood surface for $\theta_d$. If the (log-)likelihood decays quickly around the MLE, we would put more certainty on this estimate, than if the surface is essentially flat in the vicinity of the maximum. This means the estimate for $\theta_b$ appears to be more informative, and therefore reliable, than the one for $\theta_d$. Formally, this is encapsulated by the Fisher information (discussed above).

In figure 4, the observed Fisher information $FI_{obs}$ is obtained from the log-likelihood function of all parameter combinations across the parameter space. Each data point represents a summary of the curvature of the log-likelihood around the MLE for the specific parameter combination. As we can see, the existence and position of fixed points leave their mark in the Fisher information. Most importantly, the bifurcation event at $\alpha = 0$ and the unstable fixed point at $X = -\sqrt{\alpha}$ coincide with high values of the Fisher information. We note that the stable fixed point at $X = \sqrt{\alpha}$ seems to have little bearing on the Fisher information and we do not observe it in the Fisher information at $X = \sqrt{\alpha}$. This is because all initial conditions IC $> -\sqrt{\alpha}$ converge to it.

In many real applications, we know the IC and only the kinetic (bifurcation) parameter is unknown and needs to be inferred. Therefore, we shift focus on the Fisher information of the parameters but for known, specific ICs (figure 4). We chose a collection of four ICs capturing all general marginal structures found in the saddle-node bifurcation.

First, we set IC to 1.5. In this case, we observe the highest Fisher information at the bifurcation event where $\alpha = 0$; for positive $\alpha$, the Fisher information decreases when moving away from the bifurcation event. Marginalizing the Fisher information at IC $= 0$ results in a similar general shape; however, the peak of the Fisher information at 0 is even more pronounced. For IC $= -1.0$, we find again the same general structure, but the maximum's location is now shifted away from the bifurcation event to the unstable fixed point at $X = -\sqrt{\alpha}$ and the same is true for IC $= -1.5$.

## 3.3. Fisher information around the other bifurcations

In order to outline the results of the transcritical, supercritical pitchfork and subcritical pitchfork bifurcation we show the Fisher information plots in figure 5. Log-likelihood surfaces for selected parameter combinations are shown in the appendix for the transcritical (figure 7), supercritical pitchfork (figure 8) and subcritical pitchfork bifurcation (figure 9).

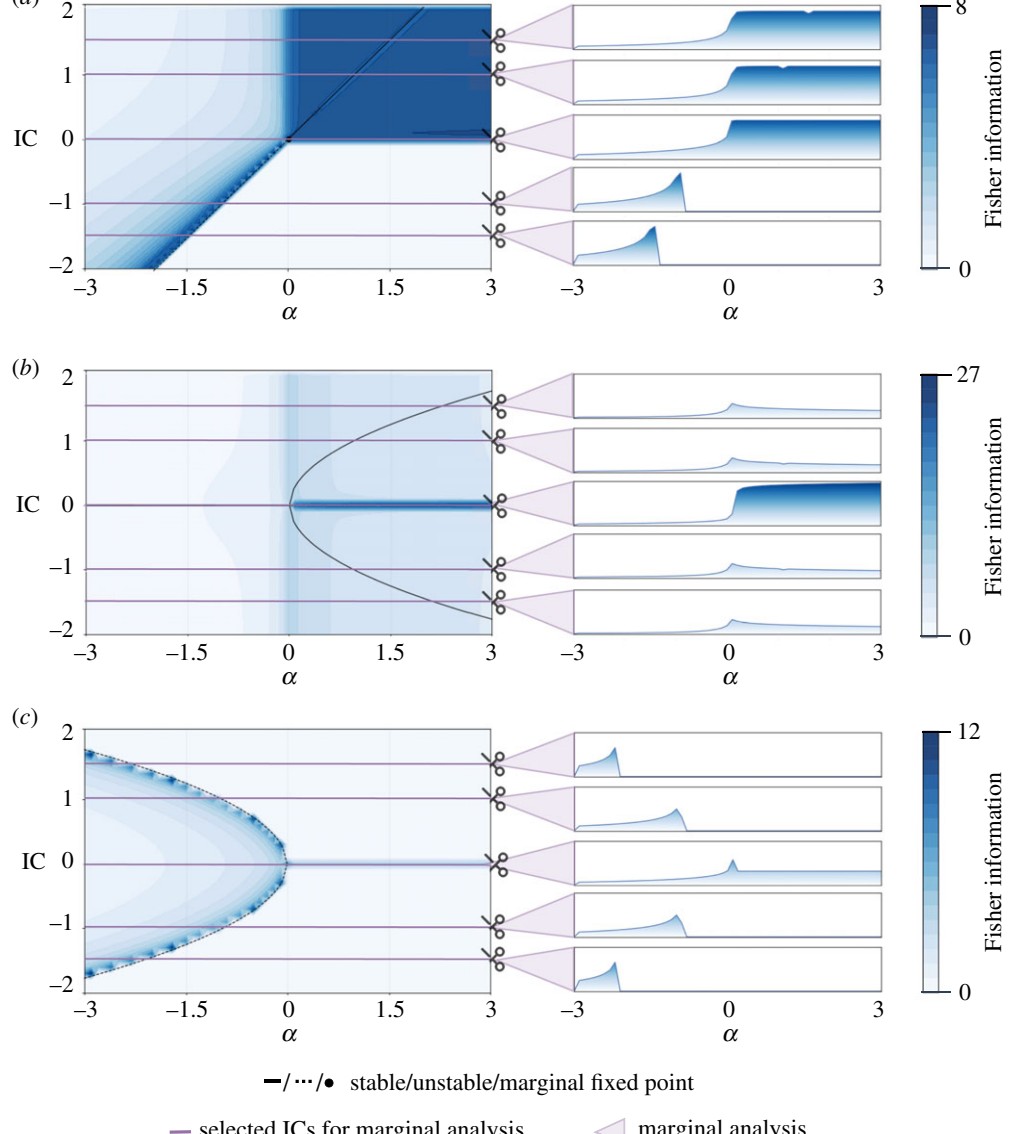

−/···/• stable/unstable/marginal fixed point

— selected ICs for marginal analysis    ◁ marginal analysis

**Figure 5.** *Quantifying information content around the transcritical, supercritical and subcritical pitchfork bifurcation via Fisher information.* For each bifurcation, the Fisher information is calculated for the whole parameter space. Each data point ($\alpha_x$, $IC_y$) represents the observed Fisher information $FI_{obs}(\theta_{x,y})$ where $\theta_{x,y} = (\alpha_x, IC_y)$. This value summarizes the curvature of the log-likelihood for the parameter combination and we use it to measure the inferability of the parameters. In order to emphasize the connection of qualitative stability properties and the inferability, the bifurcation diagrams are added. Additionally, for five IC the marginalized Fisher information with increasing $\alpha$ is presented. For all three bifurcations, we observe enriched Fisher information at the bifurcation event (transcritical: $\alpha = 0$, supercritical pitchfork: $\alpha = 0$, subcritical pitchfork: $\alpha = 0$) and along unstable fixed points (transcritical: for $\alpha < 0$ at $X_{unstable} = \alpha$ and for $\alpha > 0$ at $X_{unstable} = 0$, supercritical pitchfork: $\alpha > 0$ at $X_{unstable} = 0$, subcritical pitchfork: $\alpha < 0$ at $X_{unstable} = \pm\sqrt{|\alpha|}$ and for $\alpha > 0$ at $X_{unstable} = 0$). Equivalently to the saddle-node bifurcation and due to visualization purposes, the Fisher information is not calculatable and therefor set to 0 if the parameter combination causes the system to diverge. (*a*) Transcritical bifurcation, (*b*) supercritical pitchfork bifurcation and (*c*) subcritical pitchfork bifurcation.

### 3.3.1. Transcritical bifurcation

The transcritical bifurcation differs considerably from the saddle-node bifurcation as fixed points exist for every value of the bifurcation parameter $\alpha$. However, we still observe major differences in the parameter inferability for $\alpha < 0$ and $\alpha > 0$. For $\alpha < 0$, high Fisher information and therefore peaked likelihood surfaces are found for parameter combinations close to, but above, the unstable fixed point $X_{unstable} = \sqrt{\alpha}$. Certainty of the inference (as measured by the Fisher information) decreases as we move further away

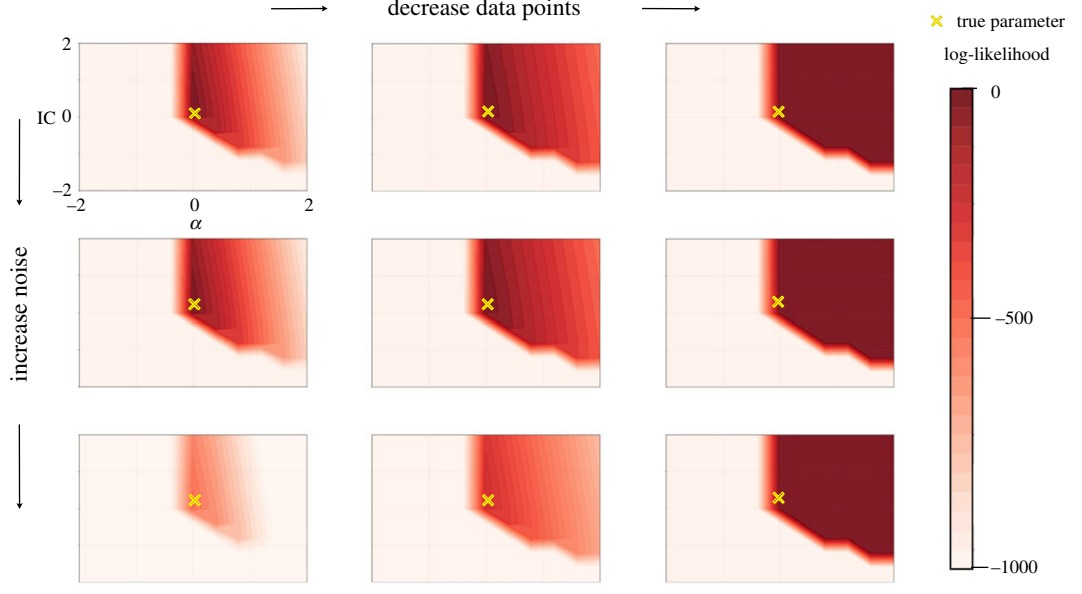

**Figure 6.** *Effect of noisy and sparse data on the shape of the log-likelihood function for the saddle-node bifurcation.* From top to bottom, we increase the noise ($\sigma = 0, 0.1, 1.0$) and from left to right, we decrease the number of observations ($M = 1000, 500, 10$). The true parameter is always kept the same. We observe that the level of noise and data density does not effect our ability of parameter inference qualitatively.

from the fixed point. In the case of $\alpha > 0$ and positive IC parameter combinations can be inferred with high certainty. The whole quadrant shows consistently high Fisher information. Interestingly, this is the only bifurcation in which the stable fixed point is clearly visible in the Fisher information (for negative $\alpha$).

### 3.3.2. Supercritical pitchfork bifurcation

For the supercritical pitchfork bifurcation, we also find at least one fixed point for every $\alpha$. Similarly to the transcritical bifurcation, we still observe variation in the inferability for positive and negative values of $\alpha$. However, we observe symmetry along the $\alpha$-axis. This is in line with our expectations, as the bifurcation diagram is symmetric, too. The highest certainty is associated with the unstable fixed point $X_{\text{stable}} = 0$. It decreases when moving away from the unstable fixed point but generally for $\alpha > 0$ the likelihood surfaces are more peaked then for negative $\alpha$. It is worth mentioning that for this bifurcation, the range of the observed Fisher information across the parameter space is significantly higher ([0, 27] for the supercritical pitchfork bifurcation versus [0, 16] for the saddle-node bifurcation, [0, 8] for the transcritical bifurcation, and [0, 12] for the subcritical pitchfork bifurcation). The curvature of the log-likelihood at negative $\alpha$, as well as for positive $\alpha$ and non-zero IC, is still sufficient for parameter inference even though they are coloured similarly to less certain predictors of other bifurcations but this is only due to the varying colour bars (see appendix A for the likelihood surfaces).

### 3.3.3. Subcritical pitchfork bifurcation

The subcritical pitchfork bifurcation is the co-dimension 1 bifurcation with the most diverging dynamics: for many IC and $\alpha$ combinations, the trajectories will not converge to a stable fixed point but diverge instead. In these areas, we cannot perform parameter inference using the MLE because we cannot determine the log-likelihood as suggested in this work. However, in areas where the solution converges to a fixed point $\left( \alpha < 0 \text{ and } -\sqrt{|\alpha|} < \text{IC} < \sqrt{|\alpha|} \right)$, parameter combinations can be inferred. Again, as for other bifurcations, the certainty associated with the inference is highest close to the unstable fixed points and moving away from the unstable fixed point results in decreasing certainty of predictions. The subcritical pitchfork bifurcation shows nicely that even if we focus on systems where the larger portion of the parameter space results in diverging dynamics, we still can use the observed Fisher information to obtain the phase structure for the underlying system.

As shown, we are able to identify the unstable fixed points at $X_{\text{unstable}} = \pm\sqrt{|\alpha|}$ for $\alpha <= 0$ and for $\alpha > 0$ at $X_{\text{unstable}} = 0$ clearly from the Fisher information.

### 3.3.4. Noisy and sparse data

In the discussion above, we have focused on idealized data that is noiseless and plentiful. Relaxing these ideal conditions does, reassuringly, not alter our findings qualitatively. As shown in figure 6, making the data sparser and adding noise still results in log-likelihood and Fisher information behaviours that recapitulate what we have described above. Thus, like in the case of the Hopf-bifurcation previously studied by Kirk *et al.* [23], we find that the qualitative hallmarks of the dynamics induced by the bifurcations persist in less than ideal, or realistic, data.

As mentioned in the introduction, many problems of inverse engineering have been solved for linear differential equations, but the difficulty exacerbates quickly for nonlinear and stochastic dynamics. Here, we have focused our analysis on nonlinear dynamics using ODE systems. We also have tested the effect of noise and sparsity in data on the parameter inference. However, using only ODE systems and adding noise to a simulation for each time point independently after solving the ODE over the complete time span, we are ameliorating the effects of noise. For explicitly stochastic dynamics, we expect deviations from the observations here, in particular for large noise. Performing an in-depth analysis for stochastic differential equations opens up a whole set of different problems, especially if we leave the low-noise regime, where our analysis would still apply.

## 4. Conclusion

We have demonstrated that the structure of a dynamical system, and the qualitative manifestations of system dynamics affect our ability to learn and infer dynamical parameters as well as initial conditions. Here, we have only considered the simplest examples of bifurcating systems, but these still showed diverse and subtle behaviour. Most importantly, the *global* dynamics can affect profoundly our ability to make *local* parameter inferences. Our discussion of results for more realistic (sparse and noisy) data suggests, that in practice, we should be able to detect the hallmarks of qualitatively different dynamics in parameter inference studies. But our analysis also suggests that some parameters will be systematically more difficult to infer than others, depending on the type of bifurcation under consideration. If a parameter is harder to infer (as e.g. measured by the Fisher information) this means that parameters in its vicinity will result in very similar system behaviour.

In real-world dynamical systems, e.g. in population and systems biology, we often expect several of these bifurcation types to be present [15]. Disentangling the interplay between global qualitative dynamics and our ability to infer parameter (or parameter combinations) will be more complicated. Some simple guidelines or heuristics, however, emerge from the present work, and these can be used as guidelines in more complicated scenarios: first, and perhaps most importantly, capturing the qualitative dynamical regime(s) qualitatively using mechanistic modelling will allow us to triage different hypotheses quickly based on even modest qualitative data; and second, the realization that exploring different initial conditions can greatly help in parameter inference; thus experimental design will be crucial in practice, as has previously been demonstrated [28,29].

Data accessibility. All calculations were carried out in the Julia language. The code is open source and available on https://github.com/LislPisl/Bifurcations, and archived within the Zenodo repository: doi:10.5281/zenodo.3403454 [30].

Authors' contributions. M.P.H.S. conceived the study and all authors took part in the further development. E.R. performed the analysis. All authors took part in writing and revising the manuscript and have read and approved it.

Competing interests. The authors declare no competing interests.

Funding. E.R. acknowledges financial support through a University of Melbourne PhD scholarship; M.P.H.S. acknowledges funding from the University of Melbourne DVCR Fund, and the Volkswagen Foundation (A122656).

Acknowledgements. We gratefully acknowledge discussions with members of the Theoretical Systems Biology group in both hemispheres. M.P.H.S. has had a long-standing interest in this problem and has initially discussed this with Erika Cule and the late Jaroslav Stark, who have both helped to shape and refine the questions addressed here.

# Appendix A

Here, we present and summarize graphically the results for the transcritical, super- and sub-cirtical pitchfork bifurcations (figures 7–9).

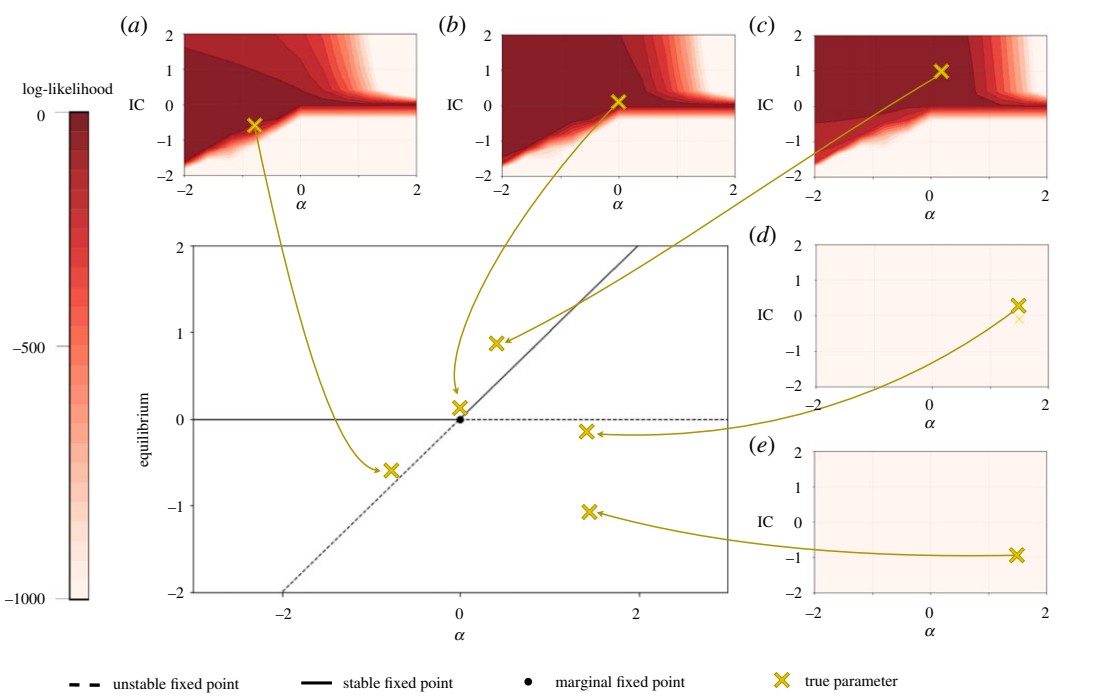

**Figure 7.** *Quantifying information content around the transcritical bifurcation via the log-likelihood.* Selected parameter combinations are highlighted in the bifurcation diagram in gold and their log-likelihood is shown in red. Parameter combinations with negative $\alpha$ and $IC > \alpha$ (a), $\alpha = 0$ and $IC > 0$ (b), or positive $\alpha$ and $IC > 0$ (c), can be inferred. However, parameter combinations below the unstable fixed points such (d) and (e), cannot be inferred.

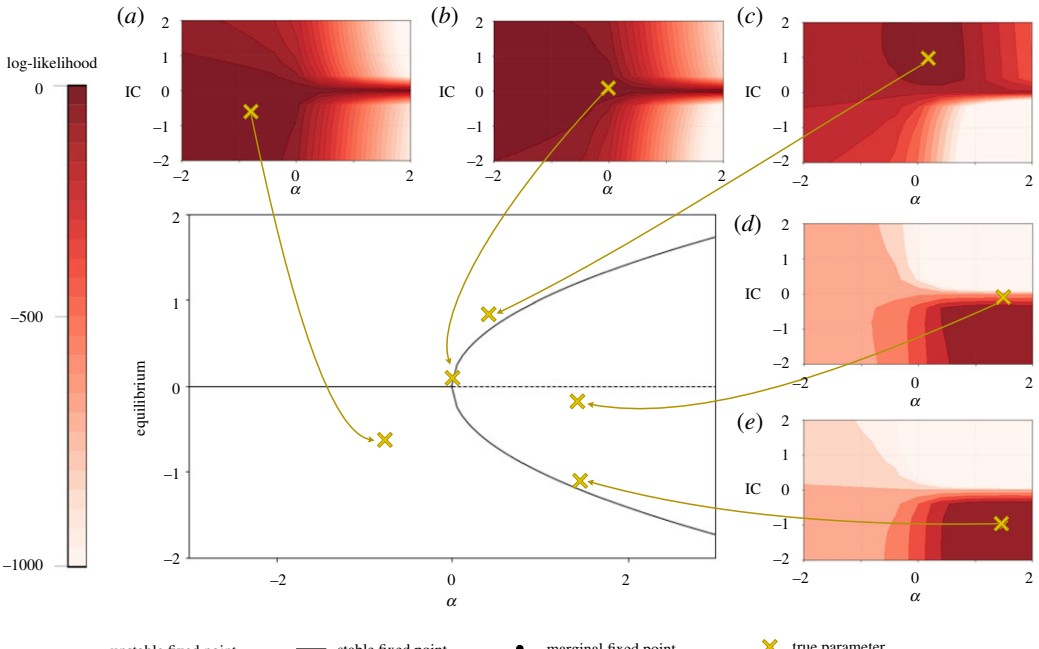

**Figure 8.** *Quantifying information content around the supercritical pitchfork bifurcation via the log-likelihood.* Selected parameter combinations are highlighted in the bifurcation diagram in gold and their log-likelihood is shown in red. For this bifurcation, parameter inference can be performed in every area of the parameter space.

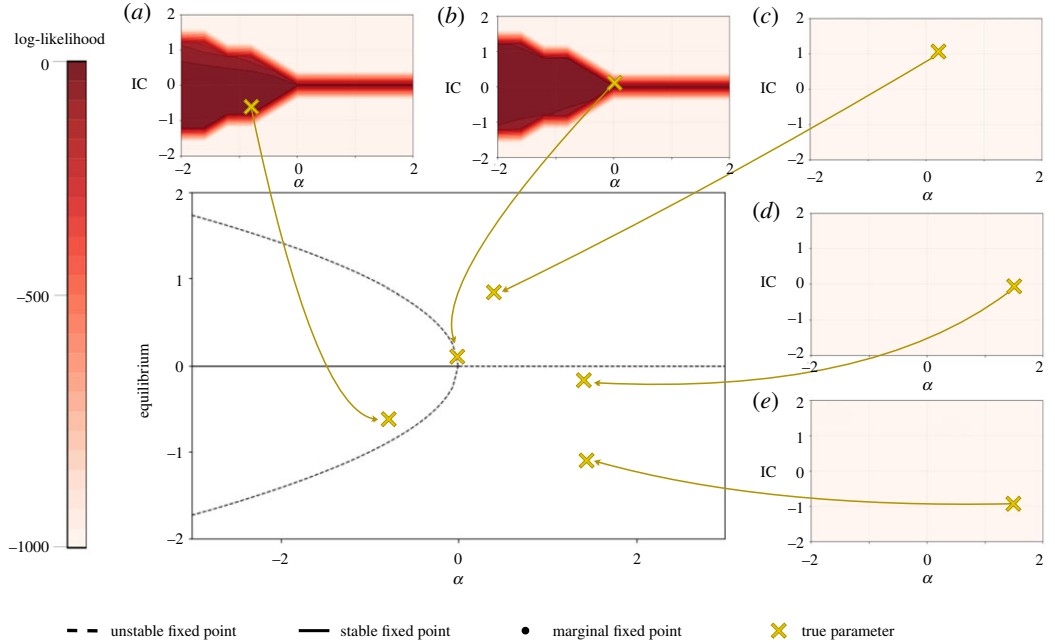

**Figure 9.** *Quantifying information content around the subcritical pitchfork bifurcation via the log-likelihood.* Selected parameter combinations are highlighted in the bifurcation diagram in gold and their log-likelihood is shown in red. Due to the unstable fixed points, parameter inference is the hardest for this bifurcation. We show only parameter combinations with negative $\alpha$ and $|IC| < \sqrt{\alpha}$ (*a*), or, $\alpha = 0$ and IC close to zero (*b*) can be inferred. Note that the true system for (*b*) is actually diverging but, due to slow divergence compared with the chosen simulated time span, the inference procedure works.

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
