## [Reviewer comments · Royal Society Open Science]

Review History

RSOS-190747.R0 (Original submission)

Review form: Reviewer 1

Is the manuscript scientifically sound in its present form?

Yes

Are the interpretations and conclusions justified by the results?

Yes

Is the language acceptable?

Yes

Do you have any ethical concerns with this paper?

No

Have you any concerns about statistical analyses in this paper?

No

Recommendation?

Accept with minor revision (please list in comments)

Comments to the Author(s)

The paper shows that parameter estimation is strongly affected by the class of dynamical system underlying the data. The authors focus in particular on simple one-parameter systems with bifurcations. The position of the system in the phase space of the model has a strong effect on parameter identifiability as quantified here by the Fisher Information. The Fisher information is often high close to bifurcation events because small parameter changes there can lead to large changes in the system behaviour and consequently a large effect on the log-likelihood of the model. The paper provides a systematic study of the Fisher information on the parameter space of different types of system exhibiting bifurcations.

I think the observations in the paper are interesting, as linking the dynamical properties of the system to parameter identifiability is of interest. The take-home message seems to be that one should consider the qualitative behaviour of the system prior to parameter estimation. However, in all but the simplest models the parameter space will be very large and therefore I'm not sure how easy it is to really map out the phase-space as is done for this simple one-dimensional system. Nevertheless, I think it is interesting to link model analysis with parameter estimation as is considered here. Previous studies looked simply at whether the parameters are inferable at all, whereas focussing on the Fisher information provides a more quantitative way to consider whether parameter estimation is feasible with finite data.

Major Comments:

- 1) Although there is no data I think it would be useful to provide the code to reproduce the figures in the paper. This should be easy to do and useful for people who wish to build on this work and get into the detail of the results.
- 2) The log-likelihood is often very far from normal or even symmetrical around the ML solution and therefore the Fisher Information doesn't fully capture the problem of parameter inference as it is really telling us about the asymptotic efficiency of parameter estimation. I think this limitation should be mentioned, that the Fisher Information is only using 2nd order information about the likelihood surface.
- 3) Related to this issue, in the context of parameter estimation in chaotic systems, for example, this does not help us much as the likelihood surface is very jagged. Therefore, this framework will not help understand every kind of dynamical system. Berliner (Journal of the American Statistical Association, 86, 1991) discusses the difficulty of interpreting the Fisher Information in such cases.
- 4) I think it should be pointed out that Bayesian parameter estimation would not require parameters to be identifiable in order to make useful inferences (e.g. about which region of the phase space we are in) and considerations about the Fisher Information are less relevant for Bayesian parameter inference. Nevertheless, I think in the Bayesian case it would still be useful to identify the phase structure of the model as one could ask the right questions, e.g. what is the posterior probability of the system being in a particular region of the phase space?

Minor comments

P6. Make it clear this is the observed Fisher Information as there are two versions.

P6. After (2.9) "we focussed on $\theta = (0, 0.1)$ " but α is studied away from zero, I didn't understand this.

Typos:

P5. "much better is than a"

P6. After (2.8) "carry about about"

P7. "sround alpha = 0"

P8. "high-values of the Fisher Information" should not be hyphenated

P10. Figure 5 caption "margnialised"

P12. "pakcage"

Review form: Reviewer 2

Is the manuscript scientifically sound in its present form?

Yes

Are the interpretations and conclusions justified by the results?

Yes

Is the language acceptable?

Yes

Do you have any ethical concerns with this paper?

No

Have you any concerns about statistical analyses in this paper?

No

Recommendation?

Accept with minor revision (please list in comments)

Comments to the Author(s)

This is a nice, short article that makes a useful contribution to an important mathematical subject that has widespread applications. The authors investigate parameter inference in a range of single-dimension, single-parameter dynamical systems with co-dimension 1 bifurcations. Multi-dimensional, multi-parameter systems that exhibit the same type of bifurcations are widespread; for example in molecular biology, ecology, and epidemiology. Parameter inference for these models is a subject of paramount importance that is still largely unsolved and therefore contributions, such as this article, are very useful.

The paper is easily readable and includes beautiful illustrations that are also fairly easy to comprehend.

I have some recommendations that I include below in order of significance:

1) There are a few surprising results that are not discussed thoroughly. For example, in supercritical pitchfork bifurcation when $a > 0$, the Fisher information is much lower at the peaks and troughs than at the unstable fixed point. However, experimental observations are often taken at the peak times. Therefore, according to the results presented here, parameter inference will often rely on these low-information (in the sense of Fisher Information) observations. I suggest that the authors highlight this and similar results, and make sure the reasons behind these observations are clearly explained.

- 2) I wish the authors have given more information regarding the effects of noise and sparsity. The general conclusion is that the results are largely unchanged. However, I would suspect some breaking points both when the observations are too sparse (e.g. in supercritical pitchfork bifurcation with $a > 0$, the observations are too sparse to observe the periodicity of the solution) or too noisy. It will be a substantial improvement if such breaking points are roughly given.
- 3) Some more details on the likelihood and the Fisher Information computation are necessary for the results to be replicable. For example, the time-points of the simulated observations. Providing the differential equations for the Fisher Information (in appendix) will also make the results more accessible.
- 4) The literature review for parameter inference on dynamical systems needs to be extended.
- 5) I suggest to clarify better the novelty compared to the most relevant publications (e.g. 22 and 23).
- 6) In section "Generating Simulated Data" where noise is discussed, I suggest a brief statement that much more detailed models of the stochasticity of dynamical systems are often necessary or useful as well as providing a few citations of relevant work.
- 6) There are a few typos that I'm sure the authors will correct in the next version.

Decision letter (RSOS-190747.R0)

09-Aug-2019

Dear Dr Stumpf,

On behalf of the Editors, I am pleased to inform you that your Manuscript RSOS-190747 entitled "Parameter inference in dynamical systems with co-dimension 1 bifurcations" has been accepted for publication in Royal Society Open Science subject to minor revision in accordance with the referee suggestions. Please find the referees' comments at the end of this email.

The reviewers and handling editors have recommended publication, but also suggest some minor revisions to your manuscript. Therefore, I invite you to respond to the comments and revise your manuscript.

- Ethics statement

- Data accessibility

If you wish to submit your supporting data or code to Dryad (<http://datadryad.org/>), or modify your current submission to dryad, please use the following link:
<http://datadryad.org/submit?journalID=RSOS&manu=RSOS-190747>

- **Competing interests**

- **Authors' contributions**

- **Acknowledgements**

- **Funding statement**

Because the schedule for publication is very tight, it is a condition of publication that you submit the revised version of your manuscript before 18-Aug-2019. Please note that the revision deadline will expire at 00.00am on this date. If you do not think you will be able to meet this date please let me know immediately.

When submitting your revised manuscript, you will be able to respond to the comments made by the referees and upload a file "Response to Referees" in "Section 6 - File Upload". You can use this to document any changes you make to the original manuscript. In order to expedite the

processing of the revised manuscript, please be as specific as possible in your response to the referees. We strongly recommend uploading two versions of your revised manuscript:

Kind regards,
Lianne Parkhouse
Editorial Coordinator
Royal Society Open Science

on behalf of Dr Robert MacKay (Associate Editor) and Mark Chaplain (Subject Editor)
openscience@royalsociety.org

Associate Editor Comments to Author (Dr Robert MacKay):

Both reviewers are positive but with minor suggestions for improvement. I recommend acceptance to revision in line with the reviewers' suggestions.

Reviewer comments to Author:

Reviewer: 1

Comments to the Author(s)

The paper shows that parameter estimation is strongly affected by the class of dynamical system underlying the data. The authors focus in particular on simple one-parameter systems with bifurcations. The position of the system in the phase space of the model has a strong effect on parameter identifiability as quantified here by the Fisher Information. The Fisher information is often high close to bifurcation events because small parameter changes there can lead to large changes in the system behaviour and consequently a large effect on the log-likelihood of the model. The paper provides a systematic study of the Fisher information on the parameter space of different types of system exhibiting bifurcations.

I think the observations in the paper are interesting, as linking the dynamical properties of the system to parameter identifiability is of interest. The take-home message seems to be that one should consider the qualitative behaviour of the system prior to parameter estimation. However, in all but the simplest models the parameter space will be very large and therefore I'm not sure how easy it is to really map out the phase-space as is done for this simple one-dimensional system. Nevertheless, I think it is interesting to link model analysis with parameter estimation as is considered here. Previous studies looked simply at whether the parameters are inferable at all, whereas focussing on the Fisher information provides a more quantitative way to consider whether parameter estimation is feasible with finite data.

Major Comments:

- 1) Although there is no data I think it would be useful to provide the code to reproduce the figures in the paper. This should be easy to do and useful for people who wish to build on this work and get into the detail of the results.
- 2) The log-likelihood is often very far from normal or even symmetrical around the ML solution and therefore the Fisher Information doesn't fully capture the problem of parameter inference as it is really telling us about the asymptotic efficiency of parameter estimation. I think this limitation should be mentioned, that the Fisher Information is only using 2nd order information about the likelihood surface.
- 3) Related to this issue, in the context of parameter estimation in chaotic systems, for example, this does not help us much as the likelihood surface is very jagged. Therefore, this framework will not help understand every kind of dynamical system. Berliner (Journal of the American Statistical Association, 86, 1991) discusses the difficulty of interpreting the Fisher Information in such cases.

4) I think it should be pointed out that Bayesian parameter estimation would not require parameters to be identifiable in order to make useful inferences (e.g. about which region of the phase space we are in) and considerations about the Fisher Information are less relevant for Bayesian parameter inference. Nevertheless, I think in the Bayesian case it would still be useful to identify the phase structure of the model as one could ask the right questions, e.g. what is the posterior probability of the system being in a particular region of the phase space?

Minor comments

P6. Make it clear this is the observed Fisher Information as there are two versions.

P6. After (2.9) "we focussed on $\theta = (0, 0.1)$ " but α is studied away from zero, I didn't understand this.

Typos:

P5. "much better is than a"

P6. After (2.8) "carry about about"

P7. "sround $\alpha = 0$ "

P8. "high-values of the Fisher Information" should not be hyphenated

P10. Figure 5 caption "margnialised"

P12. "pakcage"

Reviewer: 2

Comments to the Author(s)

This is a nice, short article that makes a useful contribution to an important mathematical subject that has widespread applications. The authors investigate parameter inference in a range of single-dimension, single-parameter dynamical systems with co-dimension 1 bifurcations. Multi-dimensional, multi-parameter systems that exhibit the same type of bifurcations are widespread; for example in molecular biology, ecology, and epidemiology. Parameter inference for these models is a subject of paramount importance that is still largely unsolved and therefore contributions, such as this article, are very useful.

The paper is easily readable and includes beautiful illustrations that are also fairly easy to comprehend.

I have some recommendations that I include below in order of significance:

1) There are a few surprising results that are not discussed thoroughly. For example, in supercritical pitchfork bifurcation when $a > 0$, the Fisher information is much lower at the peaks and troughs than at the unstable fixed point. However, experimental observations are often taken at the peak times. Therefore, according to the results presented here, parameter inference will often rely on these low-information (in the sense of Fisher Information) observations. I suggest that the authors highlight this and similar results, and make sure the reasons behind these observations are clearly explained.

2) I wish the authors have given more information regarding the effects of noise and sparsity. The general conclusion is that the results are largely unchanged. However, I would suspect some breaking points both when the observations are too sparse (e.g. in supercritical pitchfork bifurcation with $a > 0$, the observations are too sparse to observe the periodicity of the solution) or too noisy. It will be a substantial improvement if such breaking points are roughly given.

- 3) Some more details on the likelihood and the Fisher Information computation are necessary for the results to be replicable. For example, the time-points of the simulated observations. Providing the differential equations for the Fisher Information (in appendix) will also make the results more accessible.
- 4) The literature review for parameter inference on dynamical systems needs to be extended.
- 5) I suggest to clarify better the novelty compared to the most relevant publications (e.g. 22 and 23).
- 6) In section "Generating Simulated Data" where noise is discussed, I suggest a brief statement that much more detailed models of the stochasticity of dynamical systems are often necessary or useful as well as providing a few citations of relevant work.
- 6) There are a few typos that I'm sure the authors will correct in the next version.

Author's Response to Decision Letter for (RSOS-190747.R0)

See Appendix A.

Decision letter (RSOS-190747.R1)

20-Sep-2019

Dear Dr Stumpf,

I am pleased to inform you that your manuscript entitled "Parameter inference in dynamical systems with co-dimension 1 bifurcations" is now accepted for publication in Royal Society Open Science.

Kind regards,
Andrew Dunn
Royal Society Open Science Editorial Office

on behalf of Dr Robert MacKay (Associate Editor) and Mark Chaplain (Subject Editor)
openscience@royalsociety.org

Appendix A

We are grateful for the reviewers' positive assessment of our work, and we are delighted how their comments have allowed us to improve our presentation, and hopefully reach a larger audience more effectively.

Reviewer 1

Major Comments:

Comment 1:

We have modified the paragraph "DataAccessibility". It says now: "All calculations were carried out in the Julia language. The code is open source and available on <https://github.com/LisIPis/Bifurcations>."

Comment 2:

We regard the Fisher information as a useful but imperfect tool to quantify uncertainty around MLE (or MAP estimate). The limitations of the Fisher Information are now referred to in two places and we have provided some additional references regarding the use and scope of such measures.

Comment 3:

We agree entirely and have amended our discussion in light of the reviewers' comments. We would also like to thank them for the Berliner reference which we had not previously come across but which is proving useful in our ongoing work.

Comment 4:

This is a very perceptive comment (and one of the authors has argued along similar lines in the past). Again this (identifying the phase structure of dynamical systems) is part of further work, but we have added a brief Bayesian perspective.

Minor Comments:

We have done this in the revised manuscript.

Reviewer 2

Comment 1:

We had previously tried to avoid repetition but can see how this would have made some of the results harder to follow. We have now extended our discussion and elaborated on our observations and what can be learned from them.

Comment 2:

We fully understand this point. The interplay between noise and dynamics is, however, subtle. For example, some bifurcations are masked or even fully destroyed in stochastic dynamical counterparts to deterministic systems exhibiting them. We have therefore tried to focus on observational noise and experimental resolutions. Fearing that some of these results might appear obvious, if only by hindsight, we had previously opted for brevity but as the reviewer's comments make clear, our discussion is now reorganised and expanded.

Comment 3:

We are now sharing a Julia package which will enable people to reproduce our results. As the Julia syntax is closely aligned with that of other mathematical programming languages we are expecting that typical Mathematica/Matlab/Python users would straightforwardly be able to adapt the code for their purposes; or even better, explore Julia.

Comment 4:

We have done this in the revision.

Comment 5:

Our revised manuscript does this much more clearly.

Comment 6:

We have done this in the revision.